# Evaluation of the Built-Up Area Dynamics in the First Ring of Cluj-Napoca Metropolitan Area, Romania by Semi-Automatic GIS Analysis of Landsat Satellite Images

**Bogdan-Eugen Dolean** [1,*], **Ştefan Bilaşco** [1,2], **Dănuţ Petrea** [1], **Ciprian Moldovan** [1,*], **Iuliu Vescan** [1], **Sanda Roşca** [1] **and Ioan Fodorean** [1]

[1] Faculty of Geography, "Babeş-Bolyai" University, 5-7 Clinicilor Street, 400006 Cluj-Napoca, Romania; stefan.bilasco@ubbcluj.ro (S.B.); danut.petrea@ubbcluj.ro (D.P.); iuliu.vescan@ubbcluj.ro (I.V.); sanda.rosca@ubbcluj.ro (S.R.); ioan.fodorean@ubbcluj.ro (I.F.)

[2] Romanian Academy, Cluj-Napoca, Subsidiary Geography Section, 9 Republicii Street, 400015 Cluj-Napoca, Romania

[*] Correspondence: bogdan.dolean@ubbcluj.ro (B.-E.D.); ciprian.moldovan@ubbcluj.ro (C.M.); Tel.: +40743418622 (B.-E.D.); +40741258306 (C.M.)

**Abstract:** The accentuated dynamics of the real estate markets of the last 20 years, determined that a large part of the territories in the immediate vicinity of the big urban centers, to change their category of land use, in an accelerated rhythm. Most of the time, the land use changes according to the market requirements, the predominantly agricultural lands being occupied by constructions with residential or industrial functions. Identifying these changes is a difficult task due to the heterogeneity of spatial databases that come from different real estate development projects, so determining and implementing new methods to track land changes are currently highly required. This paper presents a methodologically innovative index-based approach for the rapid mapping of built-up areas, using Landsat-5, Landsat-7, and Landsat-8 satellite imagery. The approach described in this study differs from other conventional methods by the way the analysis was performed and also by the thematic indices used in the processes of built-up area delineation. The method, structured in a complex model, based on Remote Sensing and GIS techniques, can be divided into three distinct phases. The first stage is related to the pre-processing of the remote sensing data. The second stage involves the calculation of the normalized difference vegetation index (NDVI), the modified normalized difference water index (MNDWI), and the bare soil index (BI) correlated with the extraction of all areas not covered by vegetation; respectively, the elimination from the result of all areas covered by water, bare land, or uncultivated arable land. The result of this stage is represented by a distinct thematic layer that contains only built-up areas and other associated territories. The last step of the model is represented by the validation of the results, which was performed based on statistical methods and also by direct comparison with field reality, obtaining a validation coefficient which is generally above 85% for any of the methods used. The validation process shows us that by applying this method, the fast mapping of the built-up areas is significantly enhanced and the model is suitable to be implemented on a larger scale in any practical and theoretical application that aims at the rapid mapping of the built-up areas and their evolutionary modeling.

**Keywords:** remote sensing; GIS model; built-up area; mapping; urban analysis

## 1. Introduction

Urban areas have expanded at an accelerated speed during the last five decades, and rates of urban population growth are higher than the overall growth in most countries [1–3]. The accelerated development of urban areas and the anthropic pressure on the natural environment is becoming more and more prevalent recently due to population growth, resource depletion, and a modern, service-oriented economy. This has brought about many negative environmental repercussions to the world; for example, less precipitation, more dryness, higher temperatures [4,5], and increasing the probability of man-made landslides [6–9].

The conversion of the natural lands into urban areas, dominated by built-up and other associated lands, may have significant impacts on the ecosystem, which can result in negative aspects such as the urban heat island phenomenon and, on a larger scale, as global warming [10,11].

Land use changes in urban areas usually occur because of high urbanization and residential development rates, causing the suburbanization process and consequently the urban sprawl (generally known as uncontrolled urban development), which implies negative effects on human communities, such as: overcrowding, traffic congestion, open spaces loss, increasing pollution, poor housing quality, etc. [12].

Built-up area delineation is a common challenge in terms of monitoring urban development and measuring urban sprawl. This action is focused on accurately drawing the border between the built-up area of urban and rural settlements and the natural landscape. Urban environment and built-up area delimitation are subjects frequently approached in the European Union documents: EEA Report No 11/2016 regarding Urban sprawl in Europe, Urban Agenda for the EU, SOER 2015—The European environment, Europe 2020 strategy and the Environment Action Programme to 2020 [12].

Most scientific studies that analyze different aspects of urban sprawl highlights, more or less, the causal relationship between the built-up areas and the urban sprawl phenomenon [13–16]. In this sense, one of the best-documented and well-known pieces of research was elaborated by Jaeger and Schwick [17], and according to their publication, the urban sprawl degree for any given territory, results and can be quantified by combining three essential components: (1) the size of the built-up areas; (2) the spatial configuration (dispersion) of the built-up areas in the landscape; (3) the uptake of built-up area per inhabitant or job. All these measures are reunited into the Weighted Urban Proliferation (WUP) index for urban sprawl quantification; this method also used by the European Environment Agency (EEA).

Under these circumstances, the study of urban spatial expansion is very important for both urban and regional development planning, and always needs accurate and up-to-date data on built-up areas, such as: size, shape (geometry), pattern, spatial context, growth trend, and development directions. Mapping urban land in a timely and accurate manner is indispensable for quality spatial planning and rational territorial management.

Identification of the patterns of sprawl and analyses of spatial and temporal changes would help immensely in the planning for proper infrastructure facilities [18,19].

Over the past two decades, researchers have become increasingly interested in using remotely sensed imagers to address urban and suburban problems and various planning applications, such as: land use changes analysis and predictions [20–22], natural risks assessment and management [23–26], urban heat island effect [27–31], transportation system management [32], recognizing zones susceptible to pollution, and for the purpose of crop mapping in agricultural areas [33].

Approaches based on remote sensing imagery are very popular currently due to the extraordinary improvements over recent years, in terms of the quality of the data provided and also because of time and money efficiency; offering multiple mapping solutions for urban agglomerations. Remote Sensing is also used for more detailed object-based analysis by image segmentation. Recent technological advances in Geographic Information System (GIS) and image analysis offer the potential of using digital images to objectively quantify ground cover and composition of vegetation, in a repeatable and timely manner [34].

The satellite observations can provide globally consistent and repetitive measurements of the Earth's surface conditions, returning important data with different spatial, spectral, radiometric, and

temporal resolutions. They are extremely useful for monitoring the spatial distribution and growth of urban built-up areas because of their ability to provide timely and synoptic views of land cover [35,36].

Many researchers [37–45] have made use of remote sensing images to delineate the built-up areas and the majority of the approaches are based on indices or classification methods, using spectral information from satellite data. Still, the spatial and spectral variability of urban environments present fundamental challenges to obtain accurate remote sensing-based products for urban areas [41].

Land use in urban areas is very dynamic and it changes continuously, mainly due to the construction of new buildings, roads, and other infrastructures correlated with ongoing natural land take. The urban planners and decision-making actors require up-to-date land use information. Data acquisition by remote sensing sensors, integrated and further analyzed in Geographic Information System and Remote Sensing applications, has been characterized by the efficient and accurate manual extraction of the information. However, manual extraction of the information is very time consuming and requires qualified specialists; more than that, many applications do not require such a high level of detail, focusing on overviews, requiring a fast processing of numerous time series data. Therefore, to speed up this process, automatic or semiautomatic urban area delineation has become a necessity.

With the automatic delineation of settlement boundaries densification, growth and sprawl, or shrinkage, can be examined in different studies on different scales [40].

Over the years, in the field of remote sensing, scientists have developed many spectral indices for different domains and applications. For example, related to vegetation, indices such as the normalized difference vegetation index (NDVI) [30,46] and enhanced vegetation index (EVI) [47] have been developed in order to detect vegetation greenness and to rapidly map the distribution of vegetation or a variety of land surface conditions; water bodies have been extracted by using the normalized difference water index (NDWI) [48] and the modified NDWI (MNDWI) [49]; bare land have been highlighted using the normalized difference bare land index (NBLI) [42] or the normalized difference bareness index (NDBaI) [50].

A considerable number of techniques for automatically or semi-automatically mapping urban areas using satellite imagery have been developed, applied, and evaluated. These techniques can be broadly grouped into two categories: (1) those based on the classification of the input data, and (2) those based on spectral indices. This study will be focused on the second category due to the fact that the proposed methodology is focused on rapid mapping of the built-up area and, most of the time, as in the present case, this type of processing aims at the analysis of very large territories where the built surfaces are very fragmented and are characterized by a great diversity in terms of construction types, which makes it difficult and time-consuming to choose samples for supervised classification. Last but not least, it is due to the fact that, in geoinformatics, software tools are currently so developed that they allow classification based on already validated scientific criteria.

In terms of urban space, the most used indices to map built-up areas are: normalized difference built-up index (NDBI) [48], urban index (UI) [51], index-based built-up index (IBI) [36] and enhanced built-up and bareness index (EBBI) [1]. It is very important to specify that to extract specific land cover from an index resulting image, implies interactive threshold determination and repetitive attempts, which have significant implications in mapping accuracy, without the possibility of using the same values for two different images. It is commonly believed that assigning a suitable threshold value is difficult [39]. The wrong setting of thresholds will result in underestimation and overestimation of the observed land type, which leads to large errors and a poor accuracy.

The urban growth and extracted urban built-up lands in the metropolitan Washington DC area, based on a NDVI different approach for establishing urban change, filtered with a land cover classification to minimize agriculture confusion, achieved an overall accuracy of 85% [1].

The normalized difference built-up index (NDBI) to automatically map urban built-up areas was proposed by Zha et al. [45]. Therefore, the extracted built-up land information using the normalized difference built-up index is often mixed with vegetation noise, so the author had to further use the normalized difference vegetation index (NDVI) to filter out the noise. This proposed method tries to

exploit the unique spectral responses of built-up areas and other urban-related land covers but, as the authors say, their approach is unable to separate urban areas from barren and bare land, due to its complex spectral features.

The main problem with those indices is that they generate confusion between built-up areas and bare land, water bodies, and especially uncultivated arable land, due to their high complexity and similarity of spectral response patterns, especially in a mixture of pixels with heterogeneous objects [42]. For example, in the normalized difference built-up index (NDBI), index-based built-up index (IBI) and urban index (UI) images, the difference between the bare land, water bodies, uncultivated arable land, and built-up areas is very limited and the results are characterized by high uncertainties.

There are many studies that demonstrated that the built-up land class cannot be efficiently enhanced using an index constructed of original multi spectral bands because the class has a heterogeneous characteristic [49].

Thus, this study aimed to develop a better method for built-up area delineation, using free satellite imagery, without training samples or countless attempts of self-defined thresholds. In this regard, we propose a new and very useful methodology for rapid urban area mapping in a semi-automatic manner, being able to distinguish built-up and other associated lands from bare or natural areas with minimal errors. This method was successfully tested in the first ring of Cluj-Napoca's Metropolitan Area, using Landsat imagery from 2000 to 2019.

To verify the performance of the method, results have been compared with other urban-related remote sensing indices such as: normalized difference built-up index (NDBI), urban index (UI), index-based built-up index (IBI) and enhanced built-up and bare land index (EBBI), and the accuracy of the model have been also tested by multiple methods based on ground truth samples. The study results provide an alternative and more accurate remote sensing index-based method for mapping built-up and urban associated land, in a semi-automatic manner and reducing the subjectivity of the analyst.

### 1.1. Case Study

The Cluj-Napoca Metropolitan Area is located in the north-west of Romania, in the historical region of Transylvania, approximately in the center of Cluj County, at the intersection of 46°46′00.0″ north latitude parallel with the 23°36′00.0″ east longitude meridian (Figure 1).

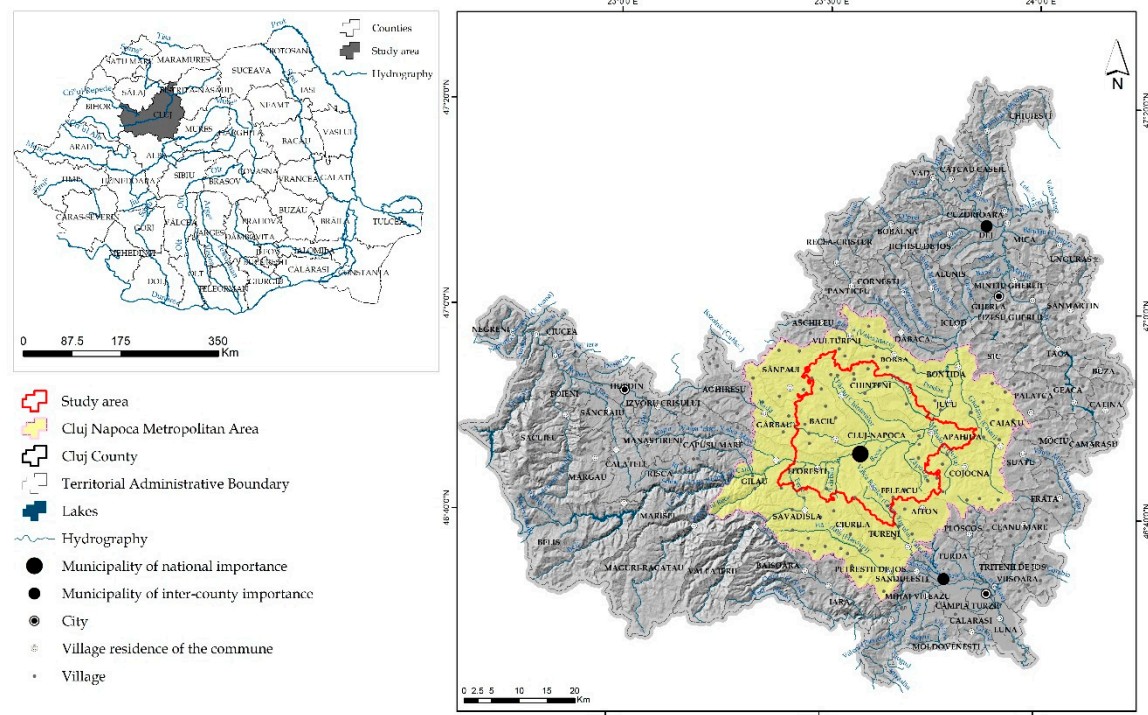

**Figure 1.** Geographic location of the study area.

From the hydro-morphological perspective, the area is located predominantly in the Someşul Mic river basin, at the intersection of the Someş Plateau, the Transylvania lowlands, and the eastern part of the Apuseni mountains (Masivul Gilău and Muntele Mare).

One of the largest in Romania, Cluj-Napoca's Metropolitan Area covers a territory of approximately 1,600 km² and includes 19 administrative units with about 100 settlements, totaling around 415,000 inhabitants (according to the General Population Census in 2011 and The Romanian National Statistics Institute); with an increase of approximately 7% compared to the total population recorded in the 2002 census.

The area is very important in terms of socio-economic and has experienced an impressive urbanization in the recent years, especially in peripheral areas around the municipality of Cluj-Napoca. The largest increase in both the population and the built-up areas, more than 90% of the total, occurred in the first ring of the metropolitan area, in the immediate neighborhoods of the municipality. Considering this, the analysis was focused on this area, administratively formed from the city of Cluj-Napoca and the following communes: Apahida, Baciu, Chinteni, Feleacu, and Floreşti.

## 2. Materials and Methods

The spatial databases used in this research are consistent with the proposed purpose of this GIS model of spatial analysis, more precisely, that of using free spatial databases and satellite imagery with high temporal extension, for the semi-automatic delineation and evaluation of the built-up area dynamics. The first stage, which aims at the rapid delimitation of the built-up areas, is based on the analysis of Landsat Level 1 satellite images by using remote-sensing spectral indices.

The remotely sensed used in this research (Table 1) are Landsat satellite images (5-Thematic Mapper; 7-Enhanced Thematic Mapper Plus and 8-Operational Land Imager & Thermal Infrared Sensor), covering a period of 19 years, between 2000 and 2019. The images used have a very good quality, a very small amount of clouds, which are generally present, outside the study area.

**Table 1.** Satellite imagery used in this study.

| Capture date | Scene | Sensor | Bands | Spatial Resolution (m) | Cloud amount (%) | RMSE (m) |
|---|---|---|---|---|---|---|
| a) 2000-08-22 | Landsat-7 | ETM+ | 8 | 30 | 1 | 2.323 |
| b) 2005-10-07 | Landsat-7 | ETM+ | 7 | 30 | 0 | 5.079 |
| c) 2010-08-26 | Landsat-5 | TM | 7 | 30 | 12 | 4.062 |
| d) 2013-08-02 | Landsat-8 | OLI&TIRS | 11 | 30 | 3.21 | 7.477 |
| e) 2016-08-26 | Landsat-8 | OLI&TIRS | 11 | 30 | 0.48 | 6.783 |
| f) 2019-07-02 | Landsat-8 | OLI&TIRS | 11 | 30 | 0.01 | 7.644 |

The Landsat Thematic Mapper (TM) sensor carried by Landsat-5 provides images in six spectral bands with a spatial resolution of 30 meters for Bands 1–5 and 7, and one thermal band (Band 6) acquired at 120-meter resolution, but re-sampled to 30-meter pixels.

The Landsat Enhanced Thematic Mapper Plus (ETM+) sensor carried by Landsat-7 satellite provides image in seven spectral bands with a spatial resolution of 30-meters for Bands 1–5 and 7. The ETM+ includes additional features in comparison with Landsat-5 TM that make it a more versatile and efficient instrument for land cover change monitoring and assessment, such as: a panchromatic band (Band 8) with a 15-meter spatial resolution; on-board, full aperture, 5% absolute radiometric calibration; thermal band (Band 6) acquired at 60-meter resolution, re-sampled to 30-meter pixels; an on-board data recorder.

Operational Land Imager (OLI) and Thermal Infrared Sensor (TIRS), two primary sensors carried by Landsat-8, generate combined images in eleven spectral bands. Seven of the eleven spectral bands are basically consistent with ETM+ found on Landsat-7. In addition, Landsat-8 provides an Ultra Blue (coastal/aerosol) and cirrus cloud detection band. Furthermore, the thermal band is acquired on two spectral channels at 100-meters (re-sampled to 30-meters) for more accurate surface temperatures [52].

### 2.1. Pre-Processing (Corrections and Calibrations)

All the images used in this study have been atmospherically and radiometrically corrected. Atmospheric correction has been done using "Dark Object Subtraction" (DOS1) model-based atmospheric correction algorithm [53,54] and the radiometric calibration has been done using the "conversion to TOA reflectance algorithm" [55], through which digital numbers (DNs-radiance) values have been converted in the "top of the atmosphere reflectance". These pre-processing methods are considered by many scientists being the most efficient and recommended for urban studies [56–58].

Given a remote sensing optical dataset, the first step is to convert the capture radiance, the raw digital number to the "top of atmosphere" values [59]. This step is necessary because it removes the cosine effect of different solar zenith angles due to the time difference between data acquisitions, it compensates for different values of the atmospheric solar irradiance arising from spectral band differences and corrects for the variation in the Earth–Sun distance between different data acquisition dates [59].

Due to the fact that images are level 1 data products, geometric corrections have been already made by the image provider (The United States Geological Survey, European Space Agency—USGS) and, because the data meet the condition according to which the root-mean-square error (RMSE) value is less than half of the spatial resolution, no additional corrections are required.

### 2.2. Method

The main steps followed in the proposed methodology are presented in this section. The approach consists mainly in an algorithm that includes three major steps; (A) pre-processing of the remote sensing data, (B) land cover types generation based on relevant spectral indices, and (C) removal of all land use classes that are not urban related and extraction of built-up area. After the image datasets pre-processing and calibration, the next step of the algorithm is represented by extracting the urban associated land use and removal of any other type (Figure 2).

For that purpose, we chose to start the analysis from a vegetation index and not from an urban related index, due to the fact that vegetation indices are most widely used and return the most accurate results if we refer to satellite remote sensing if there is an infra-red band (NIR) available.

In the preliminary analysis, both urban and vegetation indexes were compared and for this study area, normalized difference vegetation index (NDVI) returned the best results as primary dataset.

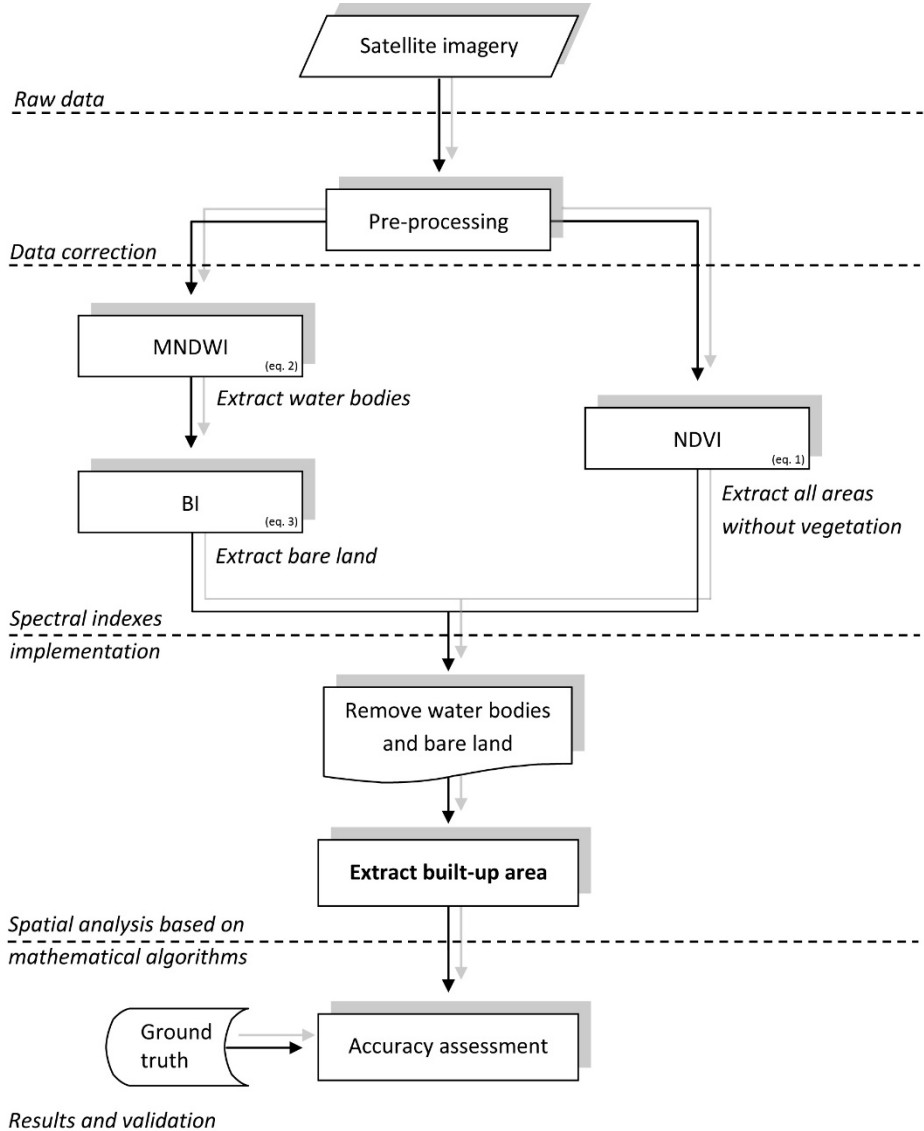

**Figure 2.** Workflow of the semi-automatic model proposed for built-up area mapping.

$$NDVI = (NIR - RED)/(NIR + RED) \tag{1}$$

There are many researchers who have used this type of approach; for example, using NDVI, SAVI (Soil Adjusted Vegetation Index) and VI (Vegetation Index) to determine manmade impervious surfaces in urban areas and achieved very good results [60–63].

The analysis proceeds from the premise according to which, after calculating the NDVI, the next step is to remove all the areas where a considerable degree of vegetation is present. For that purpose, a single threshold is required that can be adjusted depending on the degree of vegetation coverage of the analyzed area and the data acquisition season. It is very well known that NDVI returns a raster dataset with values from −1 to +1 and in general low NDVI values, below 0–0.2 means no vegetation (built-up area, roads, water, bare land, arable land, etc.), an approximate value between 0.2 to 0.5 means sparse vegetation and everything above 0.6–0.7 correspond to dense vegetation [64].

Given that the studied area does not correspond to areas with very high vegetation, in this analysis a threshold of 0.35 was chosen for the vegetation and no-vegetation areas delimitation. Of course, as we said above, in this category besides the built-up areas we will also have other types of land cover without vegetation (water bodies, bare land, arable land, etc.).

The next step of the analysis will focus on eliminating those non-urban areas using specific indexes, such as the Modified Normalized Difference Water Index (MNDWI) for water bodies and the Bare Soil Index (BI) [65,66] for bare land and arable land. The indices used for highlighting water

bodies (MNDWI), respectively; bare lands (BI) were chosen and used due to the fact that they returned the best results within the analyzed area and compared to other indices from the same category. Therefore, MNDWI best managed to highlight water bodies with variable dimensions both in area and depth and BI was best able to highlight bare and uncultivated agricultural land, in a relatively fragmented area in terms of land cover, characterized by medium density built-up areas that are often mixed with agricultural land, both cultivated and uncultivated.

$$MNDWI = (GREEN - NIR) / (GREEN + NIR) \tag{2}$$

$$BI = [(SWIR + R) - (NIR + B)] / [(SWIR + R) + (NIR + B)] \tag{3}$$

In the case of these indices, for both MNDWI and BI, the normalized values obtained will also be between −1 and 1 and the analysis proceeds from the assumption that every value above 0 means the presence of the observed features and everything below that value means the lack of it. More precisely, after the normalization of the values, from the resulting data containing values between −1 and 1, all the positive values (0 to 1) are considered as representing surfaces with the targeted characteristics and all the negative values (−1 to 0), lands with a different type of characteristics than those targeted.

Finally, the result of the analysis will be a binary raster dataset where the value 1 will mean built-up areas and other urban-related land uses (roads, railways associated land, construction sites, airports, isolated industrial structures).

## 3. Results

The implementation of remote sensing indices using Landsat Level1 satellite images provides good results in terms of built-up areas delineation, the results being thus useful in the process of spatial planning and issuing public policies for the harmonious development of the territory. Based on the integrated analysis of remote sensing indices, built-up areas for the following years 2000, 2005, 2010, 2013, 2016, and 2019 were identified, observing a strong dynamic in terms of the growth of built-up area, both for consecutive years and through the entire analysis period (Table 2).

**Table 2.** Analysis of the surfaces occupied by constructions in the 2000–2019 period.

| | Area covered by buildings. (%) | | | | | | |
|---|---|---|---|---|---|---|---|
| **TAU** | **2000** | **2005** | **2010** | **2013** | 2016 | 2019 | 2019-2000 |
| FLORESTI | 1.28 | 3.63 | 5.73 | 6.85 | 8.44 | 9.59 | 8.30 |
| BACIU | 0.43 | 0.80 | 0.99 | 1.03 | 2.39 | 3.01 | 2.58 |
| CHINTENI | 0.10 | 0.11 | 0.20 | 0.27 | 0.86 | 0.88 | 0.78 |
| APAHIDA | 2.55 | 2.85 | 3.33 | 3.38 | 6.30 | 7.91 | 5.36 |
| FELEACU | 0.02 | 0.49 | 1.92 | 2.05 | 2.46 | 2.90 | 2.88 |
| CLUJ-NAPOCA | 11.13 | 13.93 | 14.54 | 14.35 | 18.88 | 20.70 | 9.57 |

* TAU-Territorial Administrative Units (used by reference areas).

The analysis of the quantitative data obtained from the exploitation of the digital databases, resulting from the implementation of the proposed model, highlights a very high dynamic for the entire analyzed period (2000–2019), of the built territory afferent to Floreşti ATU and Apahida ATU; dynamics explained by the location of those two ATUs on the east-west development axis of Cluj-Napoca city, the axis dictated by the morphology of the place that constitutes the corridor of the Someşul Mic river.

For the municipality of Cluj-Napoca, the same accentuated dynamics can be observed, achieving an increase in the built surfaces, from 11.13% in 2000 to 20.07% in 2019, which represents an increase of almost double the amount of the built surfaces in the analyzed period.

In addition to the morphological constraints, as a negative factor for limiting the built areas, the urban restrictions imposed by the development of Avram Iancu International Airport (Cluj-Napoca), which especially affects Apahida ATU and the eastern part of Cluj-Napoca ATU, as well as the

naturally associated risks (landslides, floods); risks that are mainly manifested in the immediate vicinity of the urban area, in Apahida, Florești, and Baciu ATUs.

In the case of Florești ATU, the favorable factor of the Someșul Mic river meadow intervenes, being a relief unit which is very extensive and suitable for constructions, to which is added the relatively low price of land and the orientation of real estate developers towards areas with easy access in terms of communications and related utilities, parameters that led to an accentuated development of the locality, becoming a sleeping district of the city of Cluj-Napoca. Due to this situation, an increase of about eight times the built territory can be observed from 1. 28% at the level of 2000 to 9.59% at the level of 2019.

The small dynamics regarding the increase in the areas occupied by constructions, identified for Chinteni and Feleacu ATUs, can be explained mainly by the lack of access and utilities of public necessity, which attracted a lack of interest of large real estate developers and economic agents.

### 3.1. Built-up Area Dynamics

The visualization and evaluation of the built-up area spatial dynamics, as a result of the implementation of the proposed methodology, is possible through the process of digital mapping of the results and their correlative analysis with the main factors that dictated this dynamic.

A general analysis of the results obtained (Figure 3) clearly shows the built-up area development on the east-west axis, having as a central point the city of Cluj-Napoca. This expansion to the west of the city, which can be observed especially in the first years (2000 to 2013) of the analyzed interval, is constituted mostly on the administrative territory of Florești commune.

Starting with 2013, there is an increase in built areas in Cluj-Napoca ATU following a southerly direction but also a migration of built territories to Baciu ATU and Apahida ATU, explained by the interest of economic agents for the lands in the immediate vicinity of Cluj-Napoca ATU for the development and establishment of spaces that offer population-related services, storage spaces, logistics spaces, etc.

Starting with 2016, the more and more accentuated development of all the suitable surfaces for construction and living infrastructure stands out, observing the accentuated development and a densification of the ATUs in the immediate vicinity of the central point constituted by the city of Cluj-Napoca, especially in Florești, Apahida and Baciu ATU, a fact explained by the continuous decrease in the surfaces suitable for construction. However, the east–west development axis is kept as the main direction both for the development of built-up areas occupied by residential buildings and economic structures, the Someșul Mic corridor is still being considered the main axis of future development of the Cluj-Napoca Metropolitan Area.

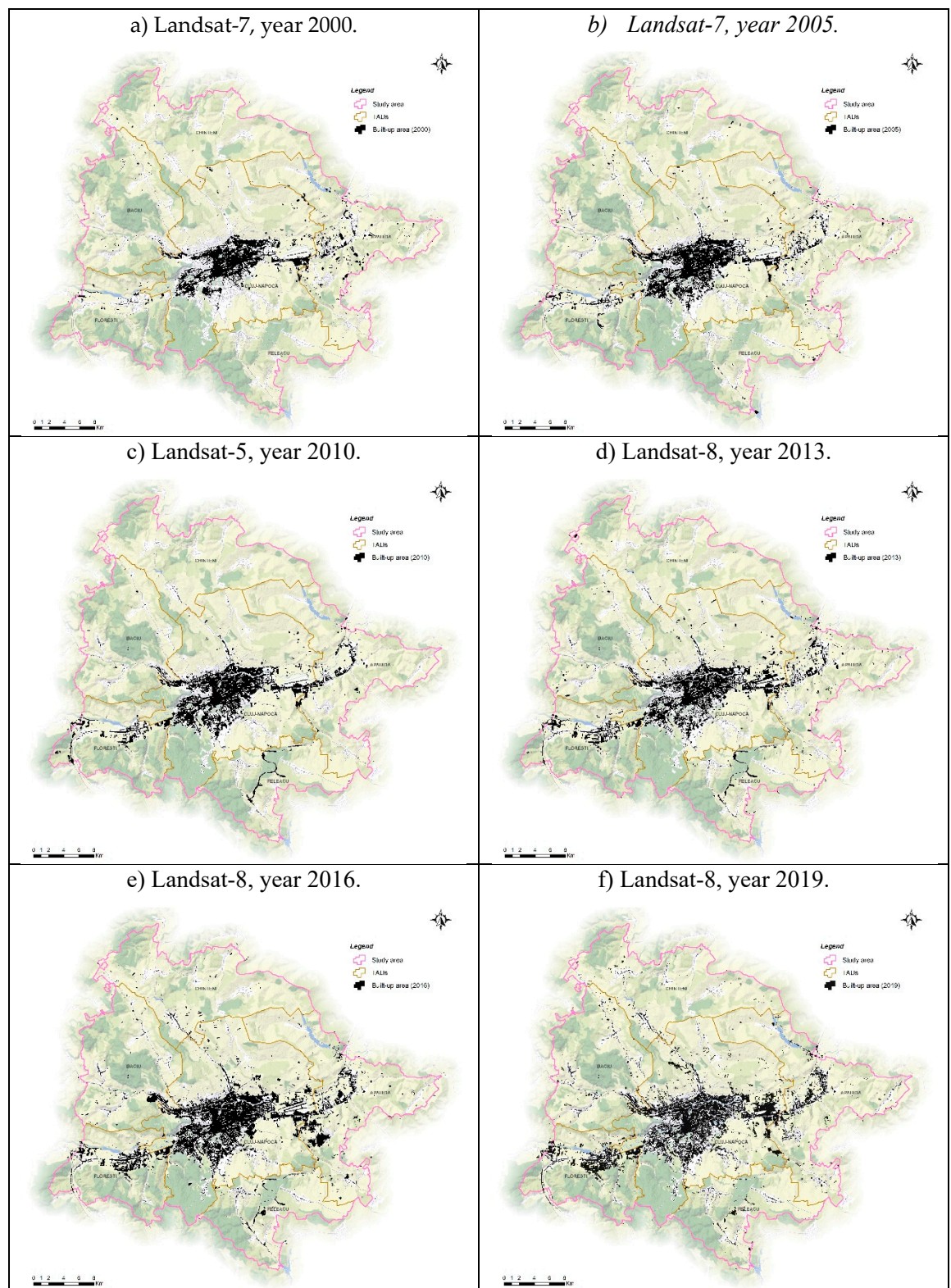

**Figure 3.** Results of the analysis based on proposed method.

*3.2. Comparison with Urban Related Indices*

The methodology proposed in the present study is based on the integrated analysis of several remote sensing indices. Considering the fact that, for the digital mapping of the built surfaces, based on the analysis of satellite images, there are a multitude of remote sensing indices applicable and

applied for various research topics or territorial analysis purposes, we consider that a comparative analysis of results is necessary to validate the obtained results.

All the results have been quantitatively and qualitatively compared to other urban related remote sensing indices, such as:

$$NDBI = (SWIR1 - NIR) / (SWIR1 + NIR) \tag{4}$$

$$UI = (SWIR2 - NIR) / (SWIR2 + NIR) \tag{5}$$

$$EBBI = (SWIR1 - NIR) / (10\sqrt{} \ (SWIR1 + TIR) \tag{6}$$

$$IBI = \frac{2SWIR1/(SWIR1+NIR) - [NIR/(NIR+Red) + Green/(Green+SWIR1)]}{2SWIR1/(SWIR1+NIR) + [NIR/(NIR+Red) + Green/(Green+SWIR1)]} \tag{7}$$

As we outlined above, most of the remote sensing indices related with built-up area extraction have a lot of noise in the results and erroneously include various land classes, such as bare land, uncultivated agricultural land, water bodies, etc. By comparing the results of NDBI, UI, EBBI and IBI (Figure 4) with the results obtained using the proposed methodology, we noticed a significant improvement in the delimitation of built-up areas with less noise and errors present (Figure 3).

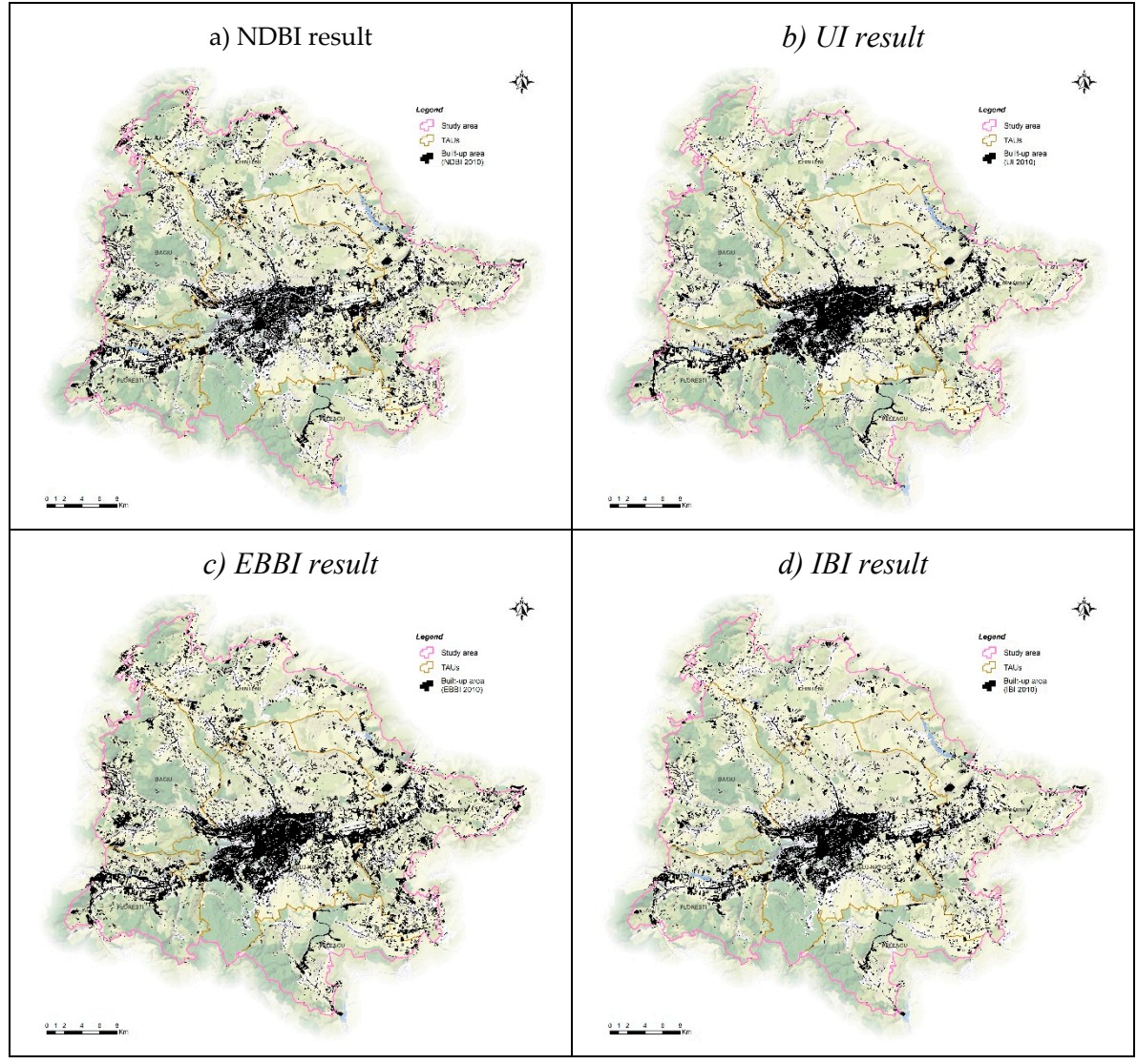

**Figure 4.** Results obtained using other urban related indices, calculated on Landsat-5 image, acquired in 2010.

*3.3. Accuracy Assessment*

The accuracy assessment was made by multiple quantitative methods so that the degree of trust can be clearly determined. The first method used was the traditional confusion (error) matrix [57], with the help of 500 randomly generated validation points, evaluated and codified accordingly for each year in question, based on a set of aerial imagery (Ortho-photos acquired in 2005 from Agency of Payments and Interventions for Agriculture—APIA, 2010 and 2012 from National Agency for Cadastre and Real Estate Advertising-ANCPI) and high-resolution images extracted from Google and Bing servers, covering the study area.

By this method, the overall accuracy is calculated by the ratio between all correct predictions and the total number of the validation samples. The Kappa coefficient which is a measure of how the classification results compare to values assigned by chance, is also determined from the confusion matrix (Table 3).

**Table 3.** Accuracy assessment and validation based on confusion matrix method.

| Year | Overall Acc. (%) | Kappa Coefficient | Producer accuracy (%) | | User accuracy (%) | |
|---|---|---|---|---|---|---|
| | | | Built-Up | Other | Built-Up | Other |
| **2000** | **88.40** | 61.15 | 94.42 | 63.15 | 82.76 | 85.42 |
| 2005 | 88.00 | 72.80 | 96.86 | 72.53 | 92.96 | 86.03 |
| 2010 | 85.20 | 70.40 | 98.80 | 71.60 | 98.35 | 77.67 |
| 2013 | 84.60 | 67.63 | 94.43 | 71.36 | 90.48 | 81.63 |
| 2016 | 87.60 | 72.23 | 93.77 | 76.54 | 87.26 | 87.76 |
| 2019 | 86.00 | 70.67 | 94.77 | 74.18 | 91.33 | 83.18 |
| Average | 86.63 | 69.15 | 95.51 | 71.56 | 90.52 | 83.62 |

As can be seen in the table above, the multiannual overall accuracy in this particular area is above 85%, what we consider a good result, taking into account the heterogeneity of the territory in question.

The second method initially involved a database creation with as many buildings markers as possible, corresponding to each year. For this step, multiple sources ware used: numerous GPS points of existing buildings, ground footprint of many buildings, measured in the field in topographic surveys (1996–2005), extracted from various urban development plans, or extracted from Open Street Map servers.

The database created has been completed and corrected manually, based on high-resolution images. Thereafter, all the buildings footprints were converted into points and from a total database of more than 65,000 points, 1000 ground-truth sample points have been randomly selected for each year in question and introduced in the validation process (Figure 5).

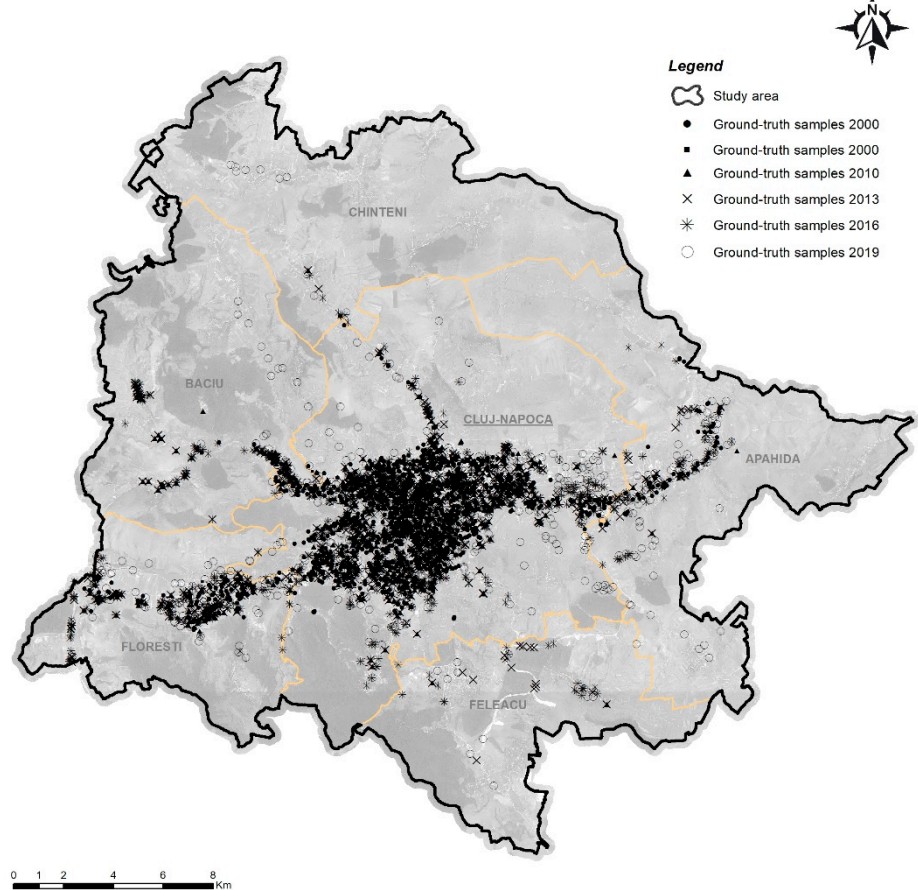

**Figure 5.** Ground-truth sample points used for validation and accuracy assessment.

By this method, the overall accuracy for each year in question has been determinate by the ratio between correct prediction and total validation sample, based on a Boolean "true" or "false" prediction table (Table 4). Based on the fact that in this case, the ground-truth samples have been obtained from real buildings location, as expected, the results were slightly better than in the previous method. This is understandable, because this method only evaluates the prediction accuracy of the built-up areas, while in the matrix of confusion the prediction accuracy has involved in the final result, the evaluation of both built-up areas and all other land use classes.

**Table 4.** Accuracy assessment and validation by method number 2.

| FIG. | YEAR | NUMBER OF SAMPLE POINTS | CORRECT PREDICTED | OVERALL ACCURACY |
|------|------|------|------|------|
| a | 2000 | 1,000 | 889 | 88.90 % |
| b | 2005 | 1,000 | 875 | 87.50 % |
| c | 2010 | 1,000 | 954 | 95.40 % |
| d | 2013 | 1,000 | 914 | 91.40 % |
| e | 2016 | 1,000 | 932 | 93.20 % |
| f | 2019 | 1,000 | 926 | 92.60 % |

The third method of validation implied a comparison between built-up area values obtained by this approach. This step is mandatory and necessary in all the GIS modeling analysis so that the final results are applicable in the specialized practice [67–70], with real statistical area values, obtained either from competent institutions, independent sources or by supervised image classification (Table 5).

**Table 5.** Accuracy assessment and validation by method number 3.

| No. | Source of data | Built-up area (ha.) | | | | | |
|---|---|---|---|---|---|---|---|
| | | 2000 | 2005–2006 | 2010 | 2012–2013 | 2015-2016 | 2018–2019 |
| 1 | I.N.S.S.E. | NO DATA | NO DATA | 5165 | 5177 | NO DATA | NO DATA |
| 2 | E.E.A. Urban Atlas | NO DATA | 5378.3 | NO DATA | 6253.44 | NO DATA | 6809.90 |
| 3 | Copernicus Urban Databases (ESM, Imperviousness) | NO DATA | 4,705.92 | 5,501.44 | 5,597.32 | 5,724.36 | NO DATA |
| 4 | Image classification | 4857.43 | 5593.38 | 6301 | 6363.93 | 6491.38 | 6731.22 |
| 5 | Mean value | 4857.43 | 5221.9 | 5655.81 | 5847.92 | 6107.87 | 6153.81 |
| 6 | Study method | 4482.13 | 5211.63 | 6246.61 | 6392.52 | 6562.35 | 6836.84 |

Even in this case, the evaluation returned relatively good results, with a predictive error ranging between ±10%–15%, and it is noteworthy that the upward trend of the built-up area expansion can be easily observed (Figure 6). This is very important when we want to quickly evaluate and map the expansion of built environments.

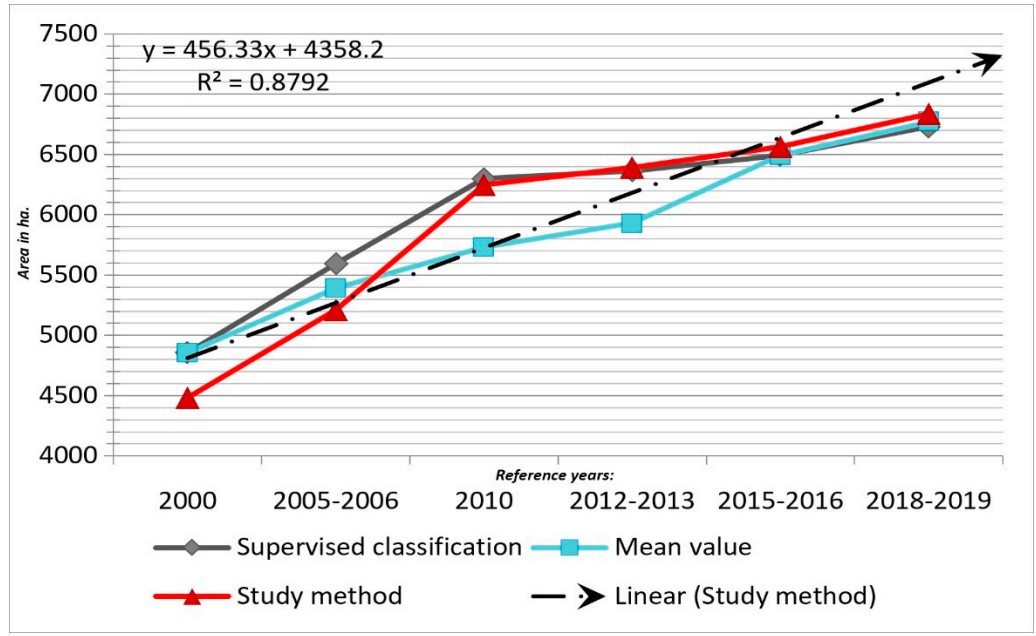

**Figure 6.** Graph representing the comparison of built-up area values (hectares) between supervised classification, mean value of other sources, and study method.

Despite the fact that the accuracy assessment returned good results, in the analysis we identified some limitations of the method. Firstly, in some cases there is a false positive prediction related to arable land uncovered by vegetation. Additionally, many constructions or dumping sites have been misidentified as built areas and in a territory where constructions are continuously present on extended surfaces, this can be an issue.

## 4. Discussions

In this study, a new approach has been proposed in order to automatically delineate the built-up area in a way that can reduce the data redundancy and spectral confusion between land-use classes.

We believe that a fast and automatic method for built-up area mapping is very useful currently, and has important potential applications, especially in the activity sphere of territorial planning and regional development.

For example, the resulting geometry and data can be very useful in terms of assessing the development potential of a territory, and multi-temporal observations are very useful for development directions, trend, and patterns analysis.

The large-scale analysis of the results of the proposed model highlights the dynamics of areas occupied by construction, dynamics that can be traced to specific periods that overlap with the main periods of "real estate boom" and accentuated economic development in the suburbs of Cluj-Napoca.

Real estate expansion is generally achieved despite the geomorphological constraints of the land, so if in the period 2000–2013, the main lands on which residential constructions were made were those from the Someșul Mic meadow, relatively flat lands and without too many geomorphological restrictions (Figure 7c,f); starting with 2016, the constructions extended in the hilly areas (Figure 7a,b, d, e), areas characterized by instability in terms of slope dynamics, requiring the implementation of complex stabilization techniques, which also involved a considerable increase in prices, affecting the entire real estate market. These observable approaches in the behavior of real estate developers are observed both in the case of residential developments and those aimed at economic projects (logistics, warehousing, retail, etc.).

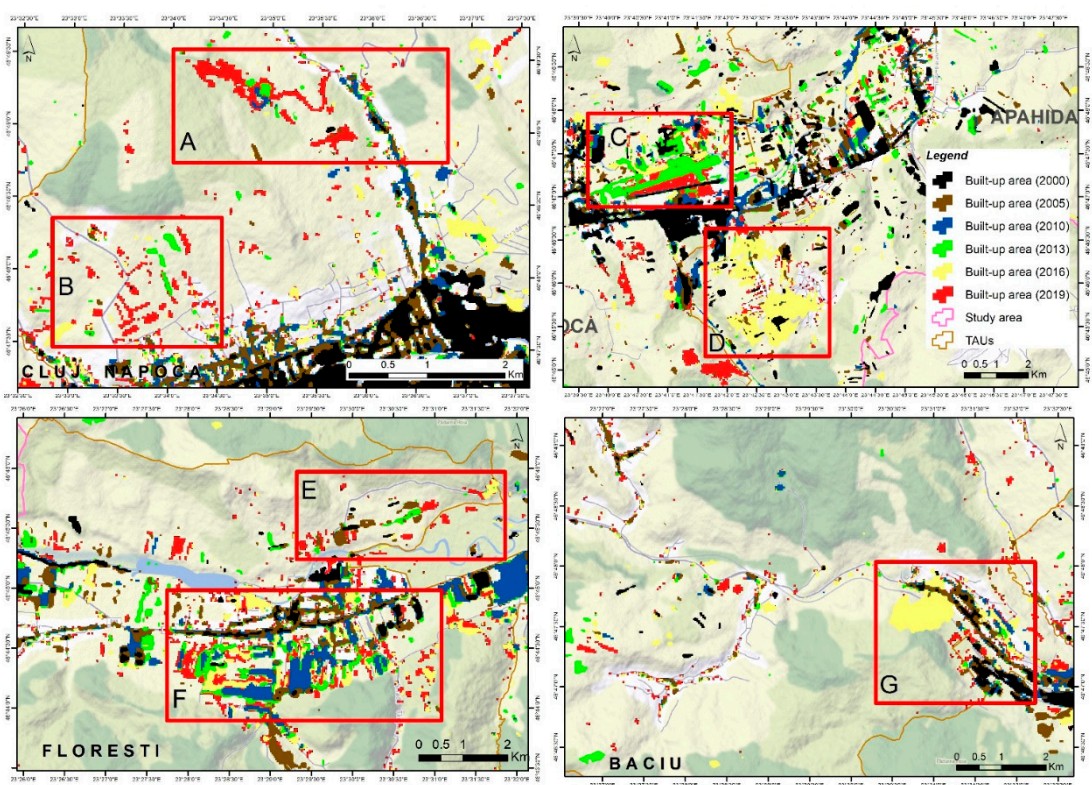

**Figure 7.** Built-up area dynamics in the period between 2000 and 2019.

The analysis of the territorial development from the perspective of the increase in the built surfaces and the associated dynamics can be followed very well in the case of the built hearth of Florești locality, it being the area with the highest dynamics in terms of built land in the entire area analyzed. Therefore, it is observed that a first phase of development, the real estate boom period that took place between 2005 and 2010, followed by the period between 2010 and 2016, a post-economic crisis period, with a development of constructions mainly in the areas without risk from a geomorphological and hydrological point of view and with a minimum cost to build, in the meadow of the Someșul Mic river (Figure 7f). This stage was followed by a new phase starting with 2016, in which the built-up areas migrated to areas with morphological instability and higher construction

cost (Figure 7e), due to the overloading of flat surfaces with constructions in period 2005–2016, which caused major problems in terms of spatial planning and urbanism.

Despite this, results illustrate that the proposed method is a relatively accurate and reliable option for a fast and automatic mapping of the built-up area validated by multiple methods, with an overall accuracy above 85%. Additionally, the comparison of relationships between the built-up area and spatial representation obtained with the proposed method and four other remote sensing indices showed that the method can highlight the built areas with higher accuracy.

We also noticed some patterns in our approach, including the fact that, in most cases, the accuracy of the method improved as the built-up area is more prominent in territory, and also it seems that the model works and produces very good results even if the raw images are used; no significant differences have been found between raw and corrected images. It is worth mentioning that the analysis can be applied for both Landsat and Sentinel images.

Nevertheless, this proposed methodology has some limitations. Firstly, regions with high landscape heterogeneity are relatively problematic. Secondly, there is still a false positive identification on agricultural lands not covered by vegetation, which must be taken into account. Thirdly, this is a semi-automatic approach where the threshold needs to be adjusted depending on the degree of vegetation presence in the analyzed area. More than that, this method is primarily based on a spectral vegetation index, which means that the images used in analysis are seasonally dependent. Thus, for optimal results we recommend images acquired between 15 June and 15 October; a range valid for areas located in the temperate climate zone.

All in all, we believe that this approach is an important contribution to the set of methods and practices with which Remote Sensing and GIS operate. We also consider that the research was a real success, but we'll also continue to seek new ways for methodological improvement.

## 5. Conclusions

This study was focused on the development of a methodology for rapid delimitation of built-up areas, based on Landsat satellite images, that are free of charge and offers global coverage and multi-temporal availability, free of charge. At the same time, one of the main objectives was to explore the possibilities of using such a model for the analysis of relatively large territories, characterized by a great variability in terms of land use/coverage, which causes a higher degree of difficulty in terms of analyzing territories based on the spectral response.

According to the mentioned goals, we consider that we have fulfilled the proposed objective, developing a methodology with relatively good results, obtaining an overall validation score of over 85% for both all the images analyzed, and methods used. As the study presented, the methodology can be successfully applied in both urban and rural areas; moreover, it can be successfully applied in the areas where the territory is in full urbanization process, in which case the urban environment expands and intertwines with the rural environment.

It is also worth mentioning that this type of analysis can provide fast mapping results for large areas, and due to the great temporal availability of Landsat satellite images, it can be used as a very useful tool for comparative analysis and evaluation of territorial dynamics from a multi-temporal perspective.

The results clearly show that through such a methodology, based on a "via negativa" approach, it is possible to extract and map elements such as built-up areas, starting from the analysis of satellite images using vegetation indices and exploiting the lack of the presence of untargeted features.

Towards the end, although this topic is frequently approached in the literature, the interest over time and the fact that the topic is still more than relevant, validates the importance of such studies. Therefore, we believe that this research makes a significant contribution to the field and aligns perfectly with similar scientific studies [36,42,45,50,65,71,72], adding value in terms of developing new practical tools for spatial analysis, management, and planning.

As for the future work, this will involve further testing of the methodology presented, in multiple areas with various environmental conditions and different characteristics in terms of land

use/land cover, aiming at a better calibration of the model and at the same time the improvement of the implementation capabilities for direct use as a tool in dedicated GIS software.

**Author Contributions:** Conceptualization, B.-E.D. and S.B.; methodology B.-E.D. and D.P.; software C.M. and B.-E.D.; validation S.R., B.-E.D. and S.B.; formal analysis I.V.; investigation B.-E.D. and S.B.; resources, I.V. and I.F.; writing—original draft preparation B.-E.D. and S.R.; writing—review and editing B.-E.D. and S.R.; visualization B.-E.D. and C.M.; supervision D.P. and B.S. All authors have read and agreed to the published version of the manuscript.

**Funding:** This research received no external funding

**Acknowledgments:** The present work has received financial support through the project: Entrepreneurship for innovation through doctoral and postdoctoral research, POCU/360/6/13/123886 co-financed by the European Social Fund, through the Operational Program for Human Capital 2014-2020. All authors have an equal contribution to this paper. All authors contributed equally to this work.

**Conflicts of Interest:** The authors declare that there are no conflicts of interest related to this article.

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
