# Peer review of "Evaluation of the Built-Up Area Dynamics in the First Ring of Cluj-Napoca Metropolitan Area, Romania by Semi-Automatic GIS Analysis of Landsat Satellite Images"

_applsci, doi:10.3390/app10217722_

Round 1

Reviewer 1 Report

The manuscript entitled “Evaluation of the built-up area dynamics in the 1st ring of Cluj-Napoca Metropolitan Area, Romania by semi-automatic GIS analysis of Landsat satellite  images” presents an interesting study focused on the use of Landsat satellite images for mapping built-up areas.

The issues raised in the manuscript are up to date and very important because the intensive and rapid development of urban areas may bring many negative effects to the environment and human communities. Because of that, it is highly desired to take actions aiming at the delineation and monitoring of urban development. The study presented in the manuscript is also valuable in aspect of planning of regional development.

The authors mentioned that the increase in remote sensing use is observed in broad range of scientific and commercial fields. It concerns for example planning, analyses of land use, risk assessment and transportation. Nevertheless, I would enhance authors to cite also studies carried out with the use of remote sensing and satellite images in agricultural areas where it is important to recognize zones susceptible to pollution and for the purpose of crop mapping. There are some studies published by MDPI related to this concern, for example:

  • Sieczka et al. (2018), https://doi.org/10.3390/w10040399
  • Demarez et al. (2018), https://doi.org/10.3390/rs11020118

In general, the content of the manuscript is presented in an understandable way and gives new insight into mapping of urban areas, taking into consideration built-up, bare or even natural areas. Nevertheless, I think that the “Discussion” should be amended, referring to some more articles to compare main findings and to formulate conclusions that could be valid in a broader scale. In the text, I would like to see whether there are any limitations in presented research and whether the presented approach would be adopted in other regions of the world. I also recommend to add a separate chapter “Conclusions” where the authors would highlight what is the most important outcome of their work.   

Author Response

Dear Editors and Reviewers,

Thank you for your reviewers’ comments concerning our manuscript entitled “Evaluation of the built-up area dynamics in the 1st ring of Cluj-Napoca Metropolitan Area, Romania by semi-automatic GIS analysis of Landsat satellite images.”. Those comments are all valuable and very helpful for revising and improving our paper, as well as the important guiding significance to our researches. 

We have studied comments carefully and have made correction which we hope meet with approval. Revised portion are marked in red in the paper. The main corrections in the paper and the responds to the Editor’s and reviewer’s comments are as flowing:

Q1. The authors mentioned that the increase in remote sensing use is observed in broad range of scientific and commercial fields. It concerns for example planning, analyses of land use, risk assessment and transportation. Nevertheless, I would enhance authors to cite also studies carried out with the use of remote sensing and satellite images in agricultural areas where it is important to recognize zones susceptible to pollution and for the purpose of crop mapping. There are some studies published by MDPI related to this concern, for example:

Sieczka et al. (2018), https://doi.org/10.3390/w10040399

Demarez et al. (2018), https://doi.org/10.3390/rs11020118

We added some more citation in the Introduction section.

Q 2. In general, the content of the manuscript is presented in an understandable way and gives new insight into mapping of urban areas, taking into consideration built-up, bare or even natural areas. Nevertheless, I think that the “Discussion” should be amended, referring to some more articles to compare main findings and to formulate conclusions that could be valid in a broader scale. In the text, I would like to see whether there are any limitations in presented research and whether the presented approach would be adopted in other regions of the world. I also recommend to add a separate chapter “Conclusions” where the authors would highlight what is the most important outcome of their work.   

In accordance with the journal's recommendations, I have not included a chapter on conclusions in the article. The lack of conclusions is supplemented by the detailed analysis of the results in the discussion section which has been updated and supplemented with additional information.

We tried our best to improve the manuscript and made some changes in the manuscript. These changes will not influence the content and framework of the paper.

We appreciate for Editors/Reviewers’ warm work earnestly, and hope that the correction will meet with approval.

Once again, thank you very much for your comments and suggestions.

Thank you and best regards.

Yours sincerely,

Reviewer 2 Report

The paper describes a semi-automatic method to analyze the evolution of urban areas through the use of remote sensing images from satellite through the combination of specific spectral indices.

My personal opinion is positive: the work is easily understandable, adequately structured and correctly written. The conclusions, in particular, objectively summarize the strengths and limitations of the proposed method.

There are a couple of important misprints and some points that should be explained better. Here's the list:

Line 126: "This study will be focused on the second category ..." - The classification methods, unsupervised or supervised, have the great advantage of using all the information produced by the sensors, i.e. all the bands, while the indices make the most of an algebraic combination of two or three bands. The former, in particular the supervised ones, are potentially superior to the latter, especially when complex and heterogeneous spectral features have to be identified. What considerations prompted the authors to adopt the index-based method? Perhaps because it is a possible solution even with GIS software without specific classification tools?

Line 223: "... by many scientists ... the most efficient" - Can the authors provide some reference to support this? Modern literature and present remote sensing software offer more advanced solutions to carry out the radiometric calibration.

Line 250: Equation 1 is an identity. There is a sign error in the denominator: -> NIR + RED

Line 252: "There are "many" researchers ..." but just one work [54] is referenced.

Line 273: How have the authors chosen and assessed the threshold for BI and MNDWI?

Line 286 (Table): "Area covered by buildings (%)" - Percentage is relative to what? What is the reference area?

Line 290: "ATU" - What does ATU stand for?

Line 299: APAHIDA case - From table 2, between 2000 and 2010, APAHIDA shows a constant decrease of the surface occupied by constructions. Is this realistic?

Line 344: Equation 4 is an identity. There is a sign error in the denominator: --> SWIR1+NIR

Line 393: "Supervised image classification" - It would be interesting to have a reference to this analysis, considering that in Table 5 and Figure 6 the values ​​of the Built-up area estimated by the Image Classification and the indexes based method tend to be in good agreement.

Best regards

Author Response

Dear Editors and Reviewers,

Thank you for your reviewers’ comments concerning our manuscript entitled “Evaluation of the built-up area dynamics in the 1st ring of Cluj-Napoca Metropolitan Area, Romania by semi-automatic GIS analysis of Landsat satellite images.”. Those comments are all valuable and very helpful for revising and improving our paper, as well as the important guiding significance to our researches.

We have studied comments carefully and have made correction which we hope meet with approval. Revised portion are marked in red in the paper. The main corrections in the paper and the responds to the Editor’s and reviewer’s comments are as flowing:

Q1. Line 126: "This study will be focused on the second category ..." - The classification methods, unsupervised or supervised, have the great advantage of using all the information produced by the sensors, i.e. all the bands, while the indices make the most of an algebraic combination of two or three bands. The former, in particular the supervised ones, are potentially superior to the latter, especially when complex and heterogeneous spectral features have to be identified. What considerations prompted the authors to adopt the index-based method? Perhaps because it is a possible solution even with GIS software without specific classification tools?

The second method was chosen for the present study due to the fact that the area taken into analysis has a very large extension, the built territory having large fragmentation, high diversity of the type of constructions which makes the choice of samples for the supervision classification difficult and last but not least due to the fact that within the geoinformation software tools are developed that allow the classification on already validated scientific criteria.

Q2. Line 223: "... by many scientists ... the most efficient" - Can the authors provide some reference to support this? Modern literature and present remote sensing software offer more advanced solutions to carry out the radiometric calibration.

We added more citation in the text for sustain our statement.

Q3. Line 250: Equation 1 is an identity. There is a sign error in the denominator: -> NIR + RED

We make the change. Thank you for your remark.

Q4. Line 252: "There are "many" researchers ..." but just one work [54] is referenced.

We added more citation in the text for sustain our statement.

Q5. Line 273: How have the authors chosen and assessed the threshold for BI and MNDWI?

We added new information in the text:

The indices used for highlighting water bodies (MNDWI), respectively bare lands (BI) were chosen and used due to the fact that they returned the best results within the analyzed area and compared to other indices from the same category. Therefore, MNDWI best managed to highlight water bodies with variable dimensions both in area and depth and BI was best able to highlight bare and uncultivated agricultural land, in a relatively fragmented area in terms of land cover, characterized by medium density built-up areas that are often mixed with agricultural land, both cultivated and uncultivated.

Q6. Line 286 (Table): "Area covered by buildings (%)" - Percentage is relative to what? What is the reference area?

We added a food note for the table 2 through which we identified the reference unit.

Q7. Line 290: "ATU" - What does ATU stand for?

We corrected in the text TAU to TAU- Territorial Administrative Units.

Thank you for your remark.

Q8. Line 299: APAHIDA case - From table 2, between 2000 and 2010, APAHIDA shows a constant decrease of the surface occupied by constructions. Is this realistic?

Thanks for the observation. The data in Table 2 have been corrected.

Q9. Line 344: Equation 4 is an identity. There is a sign error in the denominator: --> SWIR1+NIR
We correct the formula 4. Thank you for your remark.

Q10. Line 393: "Supervised image classification" - It would be interesting to have a reference to this analysis, considering that in Table 5 and Figure 6 the values ​​of the Built-up area estimated by the Image Classification and the indexes-based method tend to be in good agreement.

We added more citation in the text for sustain our statement.

We tried our best to improve the manuscript and made some changes in the manuscript. These changes will not influence the content and framework of the paper.

We appreciate for Editors/Reviewers’ warm work earnestly, and hope that the correction will meet with approval.

Once again, thank you very much for your comments and suggestions.

Thank you and best regards.

Yours sincerely,